# Electrodeposition of Nanocrystalline Fe-P Coatings: Influence of Bath Temperature and Glycine Concentration on Structure, Mechanical and Corrosion Behavior

**Natalia Kovalska [1,2,\*], Natalia Tsyntsaru [3], Henrikas Cesiulis [3], Annet Gebert [4], Jordina Fornell [5], Eva Pellicer [5], Jordi Sort [5,6], Wolfgang Hansal [1] and Wolfgang Kautek [2,\*]**

[1]  Hirtenberger Engineered Surfaces GmbH, Leobersdorferstrasse 31-33, A-2552 Hirtenberg, Austria; wolfgang.hansal@hirtenberger.com
[2]  Department of Physical Chemistry, University of Vienna, Währinger Strasse 42, A-1090 Vienna, Austria
[3]  Department of Physical Chemistry, Vilnius University, Naugarduko str. 24, LT-03225 Vilnius, Lithuania; ashra_nt@yahoo.com (N.T.); henrikas.cesiulis@chf.vu.lt (H.C.)
[4]  Leibniz Institute for Solid State and Materials Research Dresden (IFW), Helmholtzstrasse 20, D-01069 Dresden, Germany; a.gebert@ifw-dresden.de
[5]  Department de Física, Facultat de Ciències, Universitat Autònoma de Barcelona, Campus UAB, Building Cc, E-08193 Bellaterra, Spain; jordina.fornell@uab.cat (J.F.); eva.pellicer@uab.cat (E.P.); jordi.sort@uab.es (J.S.)
[6]  Institució Catalana de Recerca i Estudis Avançats (ICREA), Pg. Lluís Companys 23, E-08010 Barcelona, Spain
[\*]  Correspondence: nataliyakovalskaya@ukr.net (N.K.); wolfgang.kautek@univie.ac.at (W.K.)

**Abstract:** A detailed electrochemical study and investigation of a Fe-P glycine bath as a function of the temperature and glycine concentrations and current density, and their resulting corrosion and mechanical behavior is presented. A low addition of glycine to the electrolyte led to a drastic increase of the P content. At low Fe-P deposition rates, heterogeneous rough deposits with morphological bumps and pores were observed. By increasing the Fe-P deposition rate, the number of pores were reduced drastically, resulting in smooth coatings. Increasing the P content led to the formation of nanocrystalline grains from an "amorphous-like" state. Coatings with higher P contents exhibited better corrosion resistance and hardening, most likely attributed to grain boundary strengthening.

**Keywords:** electrodeposition; iron-phosphorous alloys; temperature; glycine additive; corrosion; hardness; friction; wear

## 1. Introduction

Amorphous alloys can exhibit a combination of striking characteristics, like magnetic [1–3] and mechanical [4] properties, as well as corrosion resistance [5,6], owing to their single-phase nature related with the lack of structural defects like grain boundaries and dislocations. Alloying transition metals with metalloids such as P, B, C, and Si can render amorphous phases. Fe-P coatings have recently raised scientific interest in respect to various advanced applications as temporary biodegradable bone replacement material [7] and alternative Li-ion storage anode [8,9]. Fe-P alloys may also serve as a substitute for widely banned hard chromium. Additionally, the co-deposition of phosphorous, e.g., with a Ni-carbon composite [10] or with Fe [11] can enhance the catalytic activity for the hydrogen evolution reaction.

There are several methods for obtaining Fe-P materials, such as rolling of sheet alloys [1], vacuum induction melting [2,4,12], rapid quenching, and single roller melt-spinning technique [13]. Electrodeposition is a low-cost technique for producing transition metal-metalloid alloy coatings. Alloy

electrodeposition is a complex process since various experimental parameters, such as current density, temperature, and concentration of active species in the electrolyte influence the overall electrochemical deposition reaction and, in turn, the alloy composition. Therefore, one has to define the experimental parameters and the electrolyte composition to obtain the targeted alloy composition and structure.

The number of suitable electrolytes for Fe-P alloys is limited [6,14–17]. The most studied baths are based on iron sulfate and hypophosphite [14]. When deposition was performed at pH < 2.2, glycine addition led to amorphous-like alloys [15]. Under these conditions the electrolyte was stable, the alloy composition was reproducible, and the coatings were uniform, smooth, and bright [17,18]. An increase in pH, current density and glycine content in the electrolyte led to a decrease of the phosphorus content in the alloys. The addition of glycine acts as a complex-forming additive for nickel [19] and iron [20]. Even low concentrations show an influence on the quality of the iron deposits. The structural characterization of the deposited Fe-P films was performed by X-ray diffraction (XRD) measurements [15] and Mossbauer spectroscopy [3,13,21]. These investigations revealed that a P content over 14 at.% resulted in amorphous-like coatings. Property analysis of Fe-P coatings was mostly pursued in respect to magnetic applications [2,16,22].

Limited information is available about the corrosion behavior [12,23–25] and mechanical properties [26] of Fe-P coatings. In general, fundamental studies showed that high concentrations of P could have a beneficial effect on the corrosion resistance and the mechanical properties. The presence of P in bulk iron alloys can decelerate the corrosion of steels in acid media [12,27]. Fe-P alloys passivate more easily and are more corrosion resistant than P-free coatings [25,28,29]. The corrosion behavior of Fe-P coatings with low P content (up to 1 wt.%) was studied in $H_2SO_4$ and HCl [12]. Phosphine generated by the hydrogen evolution during the corrosion process decelerated the acid corrosion.

The mechanical properties of the Fe–P based alloys produced by hot powder forging showed high ductility for 0.35 wt.% P [26]. Alloys with 0.7 wt.% P showed poor ductility, but higher strength ($\sim$500 MPa).

In this work, a detailed electrochemical study of the Fe-P glycine bath at various temperatures and glycine concentrations is presented. Moreover, the corrosion resistance and the mechanical properties were investigated in relation to their phosphorous content and crystallinity.

## 2. Experimental

Fe-P alloys were electroplated from an aqueous solution consisting of analytical grade 0.7 M $FeSO_4 \cdot 7H_2O$ and 0.06 M $NaH_2PO_2 \cdot H_2O$ in deionized water. To this solution, glycine was added in different concentrations (0.11, 0.21 and 0.64 M). The pH was adjusted to 2.5 by adding drops of $H_2SO_4$ and NaOH. The plating cell was a glass beaker (0.3 L) without stirring. The electrochemical measurements were performed in a conventional three-electrode set-up using a computer controlled potentiostat/galvanostat system (IPS PGU 200148, IPS Elektroniklabor GmbH & Co. KG, Münster, Germany). A brass plate with an active surface area of 2.25 cm$^2$ was used as a working electrode for cathodic polarization curves measurements and for galvanostatic electrodeposition. A gold electrode, as more noble metal than brass, served as working electrode for cyclic voltammetry measurements in order to measure striping of the layer and prevent dissolution of the substrate. It was rinsed in 0.1 M $H_2SO_4$ before deposition. Ag/AgCl/3 M KCl ($U$ = +0.21 V vs. standard hydrogen electrode, SHE) [30] served as a reference electrode. All potential values are vs. the SHE in the context of this paper. The counter electrode was an 8 cm$^2$ diameter steel plate positioned parallel to the working electrode in order to optimize the current distribution in the cell. The distance between the anode and the cathode was 10 cm. The temperature was set to 20, 40, or 60 °C. Coatings were electrodeposited up to a thickness of 11 μm. This thickness value and current efficiencies, η, were calculated based on weighing the total coating mass and on an elemental weight analyses (EDX) of the electrodeposited alloys and Faraday's law [31,32].

The surfaces were morphologically and compositionally characterized by a scanning electron microscope (SEM, Hitachi FEG-SEM S4800, Tokyo, Japan) equipped with energy dispersive X-ray

spectroscopy (EDX) detector. The crystallographic structure of the obtained coatings is studied by X-ray diffractometry (Rigaku MiniFlex II, Cu K$\alpha$ radiation, 30 kV, 30 mA, Tokyo, Japan). The mean crystallite size was estimated based on the peak width and position using Scherrer's equation.

Electrochemical polarization (corrosion) tests were also performed in a three-electrode configuration (potentiostat: Solartron SI 1287). The coated samples contacted with a copper tape were positioned as a working electrode towards an opening at the outer bottom of a Teflon beaker with ~50 mL electrolyte volume. A mercury-mercurous electrode (MSE, $U$ = +0.64 V vs. SHE) served as a reference electrode. The counter electrode was a Pt sheet. After immersion of the sample under open circuit potential (OCP) conditions, a potential sweep was performed from $-0.15$ V vs. the individual OCP up to +1.14 V at a sweep rate of 0.5 mV·s$^{-1}$. Each measurement was repeated three times to ensure reproducibility.

The mechanical test was performed on the cross-section surfaces of the as-plated coatings using a nanoindentation instrument (Anton-Paar NHT$^2$ nanoindentation tester, Graz, Austria) equipped with a Berkovich pyramidal-shaped diamond tip under load control mode. A load of 10 mN was applied with a loading segments fixed to 30 s followed by a load holding segment of 5 s and an unloading segment of 30 s. The hardness and elastic modulus are reported as an average value of 15 indentations, performed in the middle of the layer cross-sections.

Scratch tests were carried out applying an increasing normal load with the option for lateral force measurements. The normal load was linearly swept from 0 to 50 mN along the length of the scratch (600 μm) at a scratch velocity of 10 μm·s$^{-1}$. The tests were repeated four times for each sample. The initial profile at 10 μN at the location where the scratch was performed was carried out to assess the surface morphology. The actual penetration depth of the indenter under the sample surface was estimated by comparing the indenter displacement normal to the surface during scratching with the topography of the original surface at each position along the scratch length. The scratch segment, the roughness, and the slope of the surface were considered in the calculation of the indenter penetration. After the scratch, similarly a final profile was recorded to establish the residual scratch depth.

## 3. Results and Discussion

### 3.1. Potentiodynamic Investigations

Cathodic polarization curves were measured on brass, in order to investigate Fe-P electrodeposition (Figure 1). The polarization curves were divided into two regions (by a dashed line). The reaction in the first region is mainly related to hydrogen evolution. In the second region, hydrogen evolution and Fe-P deposition coexisted. Figure 1a shows the influence of the glycine, Gly, content in the electrolyte at 20 °C. The current density waves starting at ca. $-0.5$ V represent the proton reduction without Fe-P electrodeposition. An increase of the bath temperature caused a higher hydrogen evolution (Figure 1b). When the temperature of the Fe-P electrolyte (with Gly = 0.64 M) was increased from 20 up to 60 °C, the reduction potential shifted to the anodic direction and the current density, $j$, for the metal reduction reactions increased due to a higher electron transfer rate and complex stability.

The study of the Fe-P deposition potential with varying Gly concentration was performed by cyclic voltammetry (Figure 2). The potential transients were starting from the individual OCP to the negative direction with various reversal potentials. The first appearance of the anodic stripping peak correlates to the Fe-P deposition potential that is caused by the second current density wave. Cyclic voltammetry shows that the onset of the cathodic Fe-P deposition shifts to more negative potentials with increasing Gly concentration. It should be taken into account that the current density scale of Figure 2c contrasts to that of Figure 1a,b, when comparing the cathodic current waves. Table 1 shows the average potential of the second current density wave. The addition of Gly decreased the deposition rate caused by the competing hydrogen evolution. Therefore, the high concentration of adsorbed hydrogen species (H$^+$, compare [33]) decreased the active surface area for metal deposition.

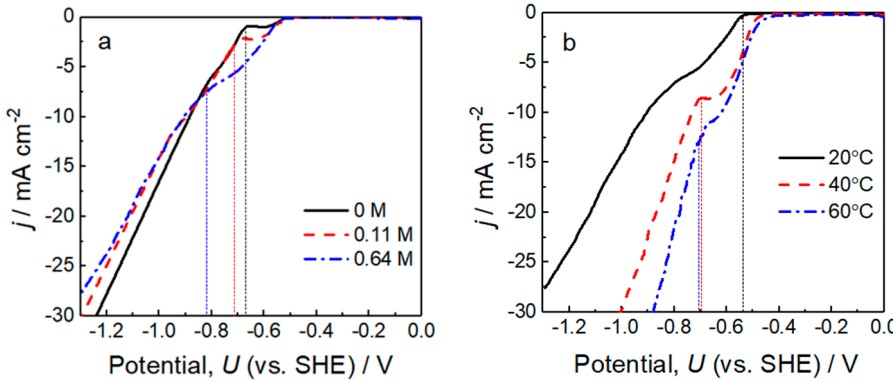

**Figure 1.** Cathodic polarisation curves of the Fe-P alloys electrodeposition depending on (**a**) the glycine concentration Gly at 20 °C and (**b**) the bath temperature at 0.64 M Gly.

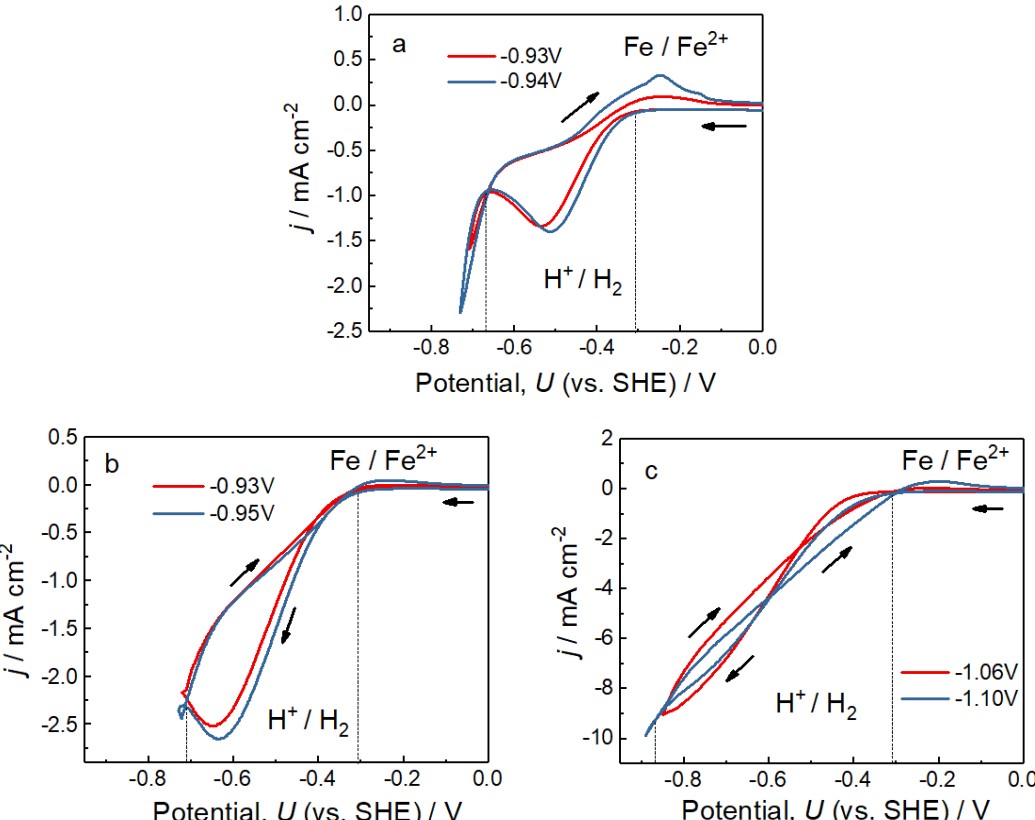

**Figure 2.** Cyclic voltammograms of the cathodic Fe-P deposition depending on the glycine concentration at 20 °C. (**a**) Glycine free; (**b**) 0.11 M Gly; and (**c**) 0.64 M Gly.

**Table 1.** Onset potential of the cathodic Fe-P deposition as a function of the glycine concentration Gly at 20 °C.

| Concentration Gly/M | Potential, $U$ (vs. SHE)/V |
|:---:|:---:|
| 0 | −0.67 |
| 0.11 | −0.71 |
| 0.21 | −0.75 |
| 0.64 | −0.85 |

### 3.2. Morphological and Structural Investigation

Galvanostatic modes dominate the technical praxis of electroplating processes. Therefore, the electrodeposition of the Fe-P coating was performed in galvanostatic mode. The deposition of the coatings was performed at 30 mA cm$^{-2}$, as an optimal current density for a high deposition rate and a homogeneous layer morphology. EDX analysis of Fe-P coatings showed a dependency of the phosphorous content in the layers on the glycine concentration (Figure 3a). A low addition of glycine to the electrolyte already had a pronounced impact on the P content. The deposition mechanism changed upon glycine adsorption [33]. Further concentration increase led to a slight P increase, due to its complexation with iron. The formed complex led to a deposition rate decreases of Fe resulting in a higher co-deposition rate of P. A temperature increase from 20 to 60 °C resulted in ~50% more P. The coatings were smooth at 40 and 60 °C, but became relatively rough at 20 °C. The oxygen content in the coatings was strongly reduced by an order of magnitude, from ca. 20 at.% to 5 at.% upon a temperature rise from 40 to 60 °C (Figure 3b). Oxygen can either derive from the deposition of oxygen species or may arise from sample oxidation upon storage in the air after electrodeposition.

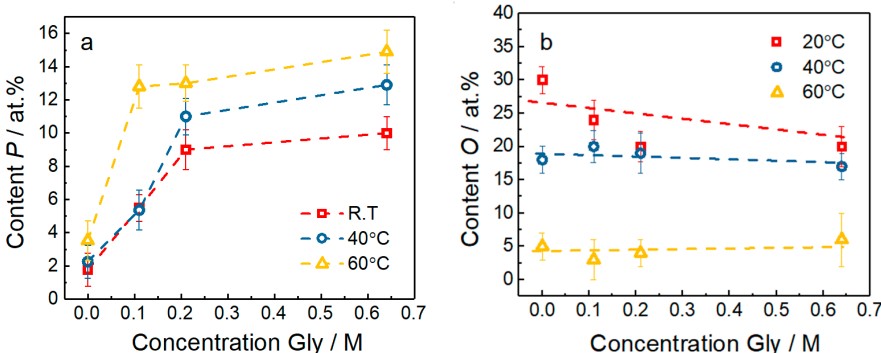

**Figure 3.** The dependency of (**a**) phosphorus and (**b**) oxygen contents of the Fe-P alloy coatings on the Gly concentration in the electrolyte at various electrolyte temperatures; $j$ = 30 mA cm$^{-2}$.

The partial current densities of the respective elements "$e$", i.e., Fe and P, $j_e$, can provide further insights into the mechanism of the Fe-P deposition. $j_e$ values were evaluated from the EDX data and the weighed coating for the respective mass according to [34]:

$$j_e = n_e \cdot \omega_e \cdot F / m_e \cdot t \qquad (1)$$

$$\omega_e = \omega t_e\% / 100 \, \Delta m / A \qquad (2)$$

where $j_e$ (mA cm$^{-2}$) is the partial current density for the corresponding element, $n_e$ the number of electrons involved in the charge transfer reaction, $\omega_e$ the mass of the element $e$ calculated from weighing and EDX analyses (g·cm$^{-2}$), $F$ the Faraday constant, $m_e$ the atomic mass of the element $e$ (g·mol$^{-1}$), and $t$ the electrodeposition time (s), $\omega t_e$ the weight content for the corresponding element analyzed with EDX, $A$ the surface area of working electrode.

The calculated $j_e$ for the Fe and P electrodeposition and the primary side reaction, the hydrogen evolution depended on the glycine concentration and the temperature (Table 2). For the sake of clarity, conditions yielding the greatest difference in oxygen content (cf. Figure 3) were used. Therefore, coatings obtained at 20 and 60 °C were compared. The partial current densities for Fe decreased with increasing the temperature, whereas that of P and the hydrogen evolution increased. An increase of Gly decreased the partial currents of Fe and P and increased the partial current for the proton discharge. The hydrogen evolution becomes the dominant reaction in the presence of a high glycine concentration at low current densities ($j$ = 15 mA·cm$^{-2}$). The occurrence of hydrogen evolution could also explain the higher amount of phosphorous in the alloy. The previous studies showed that the higher hydrogen evolution during the electrolysis process of Ni-P alloy supports the co-deposition of phosphorous [11].

**Table 2.** Influence of temperature and Gly concentration on the partial cathodic current densities, $j_e$, of the electrodeposition of Fe and P, and the hydrogen evolution (side reaction).

| $c$[Gly]/M | $T$/°C | $-j_{total}$/mA·cm$^{-2}$ | $-j_e$/mA·cm$^{-2}$ | | |
|---|---|---|---|---|---|
| | | | Fe | P | H$_2$ |
| 0.11 | 20 | 15 | 11.1 | 0.4 | 3.5 |
| | | 30 | 26.5 | 0.9 | 2.6 |
| | 60 | 15 | 9.3 | 0.7 | 5.0 |
| | | 30 | 18.9 | 1.5 | 9.5 |
| 0.64 | 20 | 15 | 8.8 | 0.9 | 5.4 |
| | | 30 | 20.6 | 2.8 | 6.6 |
| | 60 | 15 | 3.5 | 0.6 | 11.0 |
| | | 30 | 18.8 | 1.8 | 9.3 |

The results of the respective current efficiency, η, of the Fe-P alloy deposition together with the P layer content (Figure 4a) and the deposition rate (Figure 4b) are presented versus the applied current density at 60°. The chosen temperature is 60 °C, because it shows the best coating quality (roughness) and the lowest oxygen content. Current efficiency and the deposition rate were determined based on Faraday's law [31]. The phosphorous content is practically unaffected by $j$. However, the P concentration increases drastically with the glycine concentration of the bath. The influence of the Gly concentration on the η, is significant at low $j$ (glycine reduces η) and practically disappears at increased values, i.e., at 30 mA cm$^{-2}$. The deposition rate is increasing with $j$ (Figure 4b).

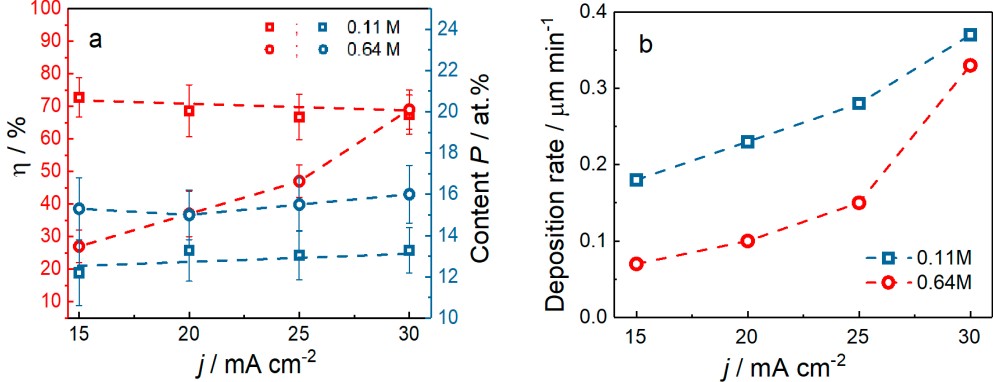

**Figure 4.** (**a**) The dependency of the current efficiency, η (grey symbols), and the phosphorous content (black symbols) in the Fe-P alloy coatings. (**b**) Dependency of the deposition rate on the cathodic current density, $j$. Variable parameter: Gly concentration. 60 °C.

The layer morphologies are very much affected by $j$ and the concentration of glycine (Figure 5). Both parameters control η (Figure 4a), a low η indicates a high proton reduction rate and a strong hydrogen gas evolution. At low glycine content, η is relatively high and practically independent of $j$. Therefore, hydrogen bubbling should be moderate at all investigated current densities. The competition between hydrogen bubble evolution and the layer growth control the morphology (Figure 5a,b). At low $j$ and low Fe-P deposition rate, inhomogeneous deposits with bumps leaving out pores of ca. 0.3 μm in diameter were observed (Figure 5a). Hydrogen bubbles generated these bumps and pores. With increased $j$ and Fe-P deposition rate, the bumps cannot form again, and the number of pores is reduced drastically (Figure 5b).

At high glycine content and low $j$, current efficiency showed <30%. The current efficiency caused vigorous hydrogen bubbling resulting in, bumps and many pores with ca. 0.1 μm in diameter (Figure 5c). An increase of $j$ caused a drastic increase of η (Figure 4a) resulting in a minimum gas evolution and as such the deposit exhibited no pores and practically no bumps (Figure 5d).

X-ray diffraction patterns of the Fe-P alloy layers grown at 60 °C showed a transition from a nanocrystalline to an amorphous-like structure; identified by peak broadening (Figure 6). An increase of the phosphorous content from 6 at.% to 16 at.% resulted in a grain size reduction from ~50 to ~6 nm ("amorphous-like") based on the analysis of the {110} reflection at 44.9°, which is in good agreement with literature [15]. Pure Fe was electrodeposited from a similar bath without hypophosphite in order to compare with Fe-P alloys. The corresponding XRD scan showed narrower peaks indicating a polycrystalline body-centered cubic Fe structure [34].

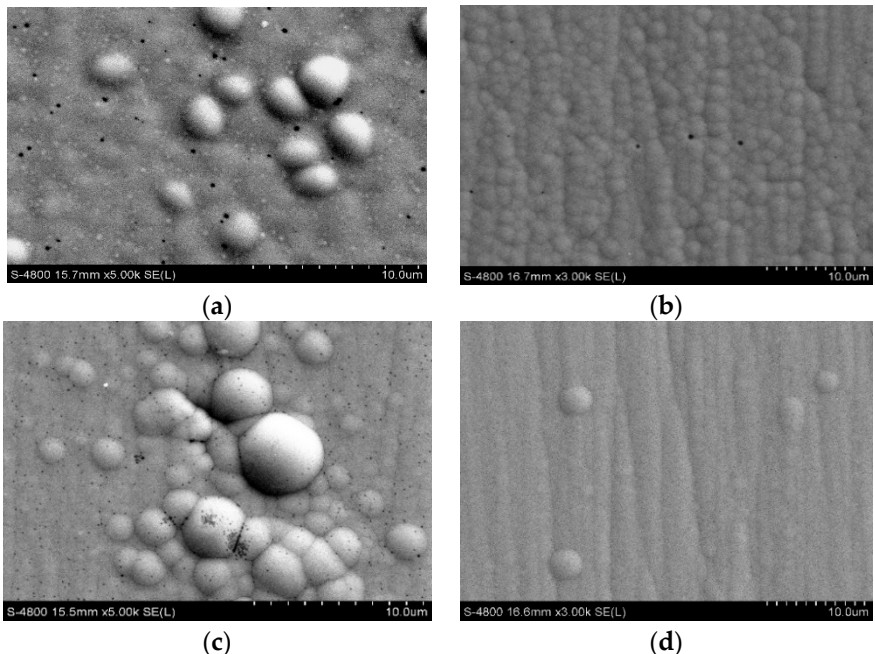

**Figure 5.** Morphology (SEM images) of Fe-P coatings electrodeposited at various Gly concentrations and current densities, *j*, at 60 °C. (**a**) Fe-12 at.% P ($j_e$ = 15 mA cm$^{-2}$, *c*[Gly] = 0.11 M); (**b**) Fe-13 at.% P ($j_e$ = 30 mA cm$^{-2}$, *c*[Gly] = 0.11 M); (**c**) Fe-14 at.% P ($j_e$ = 15 mA cm$^{-2}$, *c*[Gly] = 0.64 M); and (**d**) Fe-16 at.% P ($j_e$ = 30 mA cm$^{-2}$, *c*[Gly] = 0.64 M).

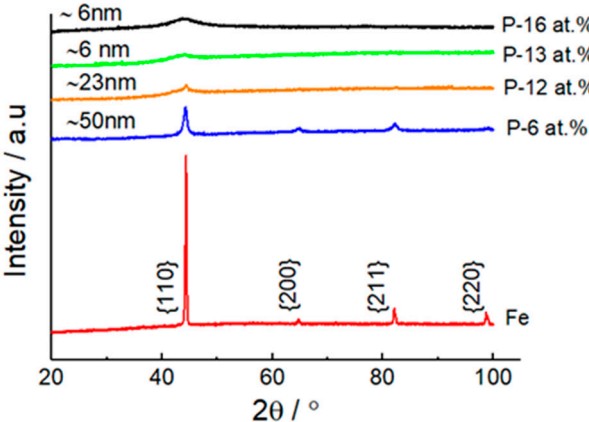

**Figure 6.** X-ray diffraction patterns of Fe-P alloys with varying P contents. The coatings were electrodeposited at 30 mA cm$^{-2}$ and 60 °C from electrolytes containing various Gly concentration: 0 M (Fe-6 at.% P), 0.11 M (Fe-12 at.% P), 0.21 M (Fe-13 at.% P), and 0.64 M (Fe-16 at.% P).

### 3.3. Corrosion Properties

In order to simulate acidic rain in an industrial environment, the corrosion properties of electrodeposited Fe-P coatings with various alloy compositions were investigated in 0.05 M H$_2$SO$_4$

(pH 1.4). The average coating thickness was around 10–12 μm. The phosphorous content of the samples were varied from 6 at.% to 16 at.% (cf. Figure 5b,d and Figure 6). The corrosion resistance $\rho_{corr}$ of a pure Fe coating and bare brass was also evaluated (Figure 7) as reference samples. The corrosion resistance is shown in Table 3 based on the corrosion potential $U_{corr}$ and the corrosion current density $j_{corr}$ which is determined by the intersection of the cathodic and anodic Tafel line extrapolations.

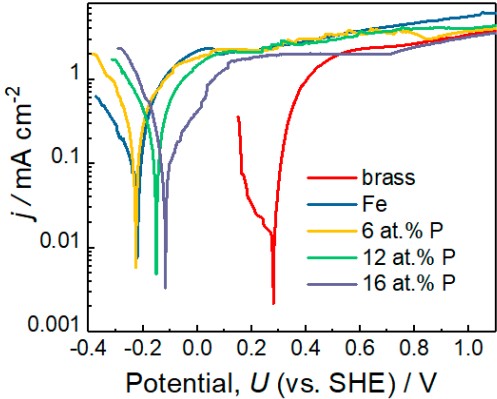

**Figure 7.** Potentiodynamic polarization curves of Fe-P coatings with various phosphorous content, compared with Fe and brass (substrate). In 0.05 M $H_2SO_4$. Fe-6 at.% P (Gly 0 M), Fe-12 at.% P (Gly 0.11 M), and Fe-16 at.% P (Gly 0.64 M).

**Table 3.** Corrosion potential ($U_{corr}$), corrosion current density ($j_{corr}$), corrosion resistance $\rho_{corr}$ of Fe-P and Fe samples.

| Phosphorous Content/at.% ($\pm1$) | $-U_{corr}$/V vs. SHE ($\pm0.5$) | $j_{corr}$/mA cm$^{-2}$ ($\pm1$) | $\rho_{corr}$/$\Omega\cdot$cm$^2$ ($\pm0.5$) |
|---|---|---|---|
| Fe | 0.22 | 0.24 | 0.49 |
| Fe-6 P | 0.20 | 0.30 | 0.62 |
| Fe-12 P | 0.15 | 0.27 | 0.59 |
| Fe-16 P | 0.12 | 0.16 | 0.67 |

Layers with a higher P content (12 at.% and 16 at.%) showed a more anodic $U_{corr}$ meaning that these alloys are more electropositive (noble) than the low alloyed and the pure Fe samples (Figure 7). The layer with the highest P content of 16 at.% showed a moderate corrosion resistance: in a potential range of ca. 0.4 V anodic of $U_{corr}$, the current density values are almost one order of magnitude lower than the values of all other samples. This is also expressed by a somewhat lower $j_{corr}$ and higher $\rho_{corr}$ (Table 3). Actually, all layers except the sample with the highest P content (16 at.%) were almost completely dissolved after the corrosion test. The coatings with <16 at.% P were more or less inhomogeneous exhibiting pores that allowed contact to the more electropositive brass electrode (Figure 5). However, the thickness of the 16 at.% P sample was moderately reduced from ~10 to ~8 μm and showed a strong crackelation (Figure 8). All curves reach an anodic current density plateau at relatively high values (>1 mA cm$^{-2}$). This may be interpreted as a diffusion-limited dissolution current or as a "pseudo-passive" behavior.

Even though pure Fe is active in the studied pH region an inhibition process [35,36] involving an adsorbed film of hypophosphite generated by the oxidation of P may play a major role:

$$P + H_2O \rightarrow H_2PO_2^- + 2H^+ + e^- \tag{3}$$

The adsorbed $H_2PO_2^-$ anion may be oxidized to $H_3PO_3$ or $H_3PO_4$, providing a P-enriched Fe-P surface. The coatings with high P content (16 at.%) were measured by EDX after the corrosion test (Table 4) to assess compositional consistency. The homogeneous 16 at.% P sample almost doubled its P content. The O content increased by a factor of four probably due to the formation of the phosphite or phosphate layer.

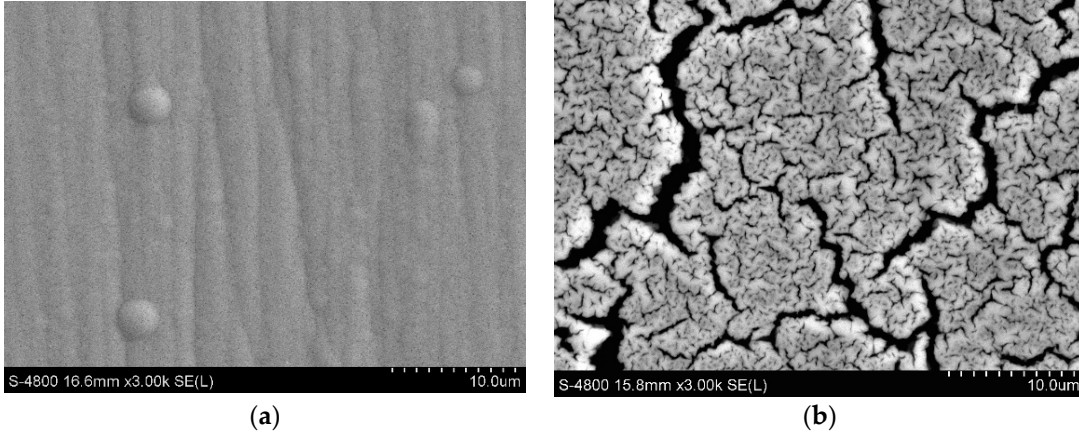

**Figure 8.** Surface morphology (SEM images) of the Fe-16 at.% P alloy before (**a**) and after (**b**) the potentiodynamic corrosion test in 0.5 M $H_2SO_4$.

**Table 4.** P and O content of Fe-P alloys before and after corrosion testing in 0.5 M $H_2SO_4$ for 1 h (comp. Figure 7). Determined by SEM-EDX.

| Elements Content/at.% ($\pm2$) | |
|---|---|
| **Before Corrosion Test** | **After Corrosion Test** |
| 6 P-<10 O | 16 P-74 O |
| 12 P-<10 O | 23 P-48 O |
| 16 P-<10 O | 24 P-41 O |

### 3.4. Mechanical Properties

The hardness, *H*, and reduced Young's modulus, $E_r$, values of the Fe-P coatings were extracted from the unloading segment of the load-displacement curves using the method of Oliver and Parr [37]. An increase of *H*, from 8.04 to 9.01 GPa, was observed as the P content increases from 6 at.% to 16 at.% (Table 5). The increase of *H* observed in the coatings from 6 at.% to 12 at.% P can be ascribed to grain boundary strengthening due to grain size reduction and to partial amorphization of the alloy as a result of P addition. In turn, the further increase of *H* observed in the coating with 16 at.% P can be mainly attributed to the amorphous-like structure of the coating with substantial amounts of P. Typically, amorphous materials exhibit larger *H* values compared to their crystalline counterparts as the conventional deformation mechanisms of crystalline materials (e.g., dislocation motion, stacking faults, grain boundary sliding) are no longer operative in amorphous alloys due to the lack of long-range order. $E_r$ decreases from 167 to 142 GPa when the P content increases from 6 at.% to 12 at.%. The variation in $E_r$ can be related to the microstructural changes (i.e., amorphization) occurring upon P enrichment.

**Table 5.** Hardness H (GPa) and Young's modulus $E_r$ (GPa) of the Fe-P coatings with P (6 at.%; 12 at.% and 16 at.%).

| P Content/at.% | *H*/GPa | $E_r$/GPa |
|---|---|---|
| Fe-6 P | 8.04 $\pm$ 0.42 | 167.0 $\pm$ 8.6 |
| Fe-12 P | 8.57 $\pm$ 0.27 | 142.3 $\pm$ 3.9 |
| Fe-16 P | 9.01 $\pm$ 0.16 | 146.6 $\pm$ 2.4 |

The wear behavior of the Fe-P coating was studied for the different investigated samples. Although similar performance is observed for the three investigated conditions, a smother friction force curve as a function of the scratch distance is observed for the Fe-16 at.% P coating when compared with the curves of the Fe-6 at.% P and Fe-12 at.% P coatings. The sharp drops observed in the friction force curves and in the friction coefficient curves (Figure 9a,b) can be ascribed to the inhomogeneous

morphology observed by SEM (Figure 5). Nonetheless, no mechanical failure is observed for any of the investigated coatings in the measured range.

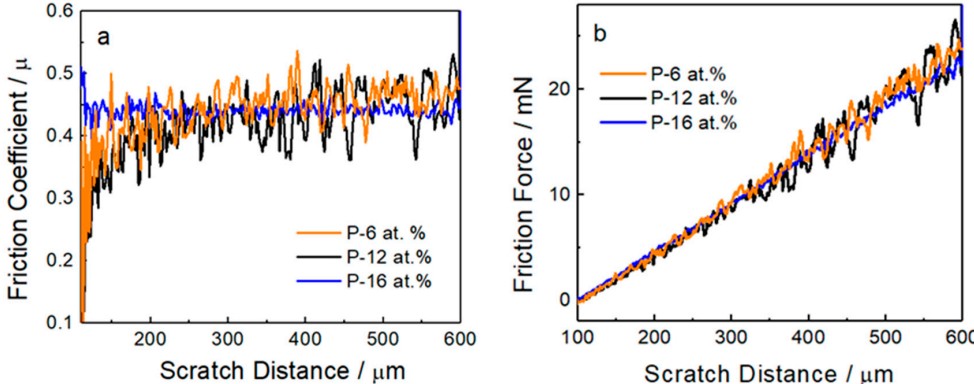

**Figure 9.** (**a**) Friction coefficient and (**b**) Friction force of measured Fe-P coatings with varying phosphorous content.

## 4. Conclusions

A detailed electrochemical study of the Fe-P glycine bath at various temperatures and glycine concentrations together with corrosion and mechanical behavior investigation was presented.

A low addition of glycine to the electrolyte of 0.11 M led to a drastic increase of the P content from ca. 4 at.% to 13 at.%. The deposition mechanism changed upon glycine adsorption. The increase of the Gly concentration decreased the partial currents of Fe and P, but increased the hydrogen evolution which supported the co-deposition of P.

The observed layer morphologies were controlled by the competition between the hydrogen bubble evolution and the layer growth. At a low Fe-P deposition rate, heterogeneous deposits with bumps and pores were observed. With an increased Fe-P deposition rate, the number of pores was reduced drastically, and smooth coatings resulted. An increase of the P content from 6 at.% to 16 at.% resulted in a grain size reduction from ~50 to ~6 nm, nanocrystalline to "amorphous-like".

An increase of the bath temperature from 20 up to 60 °C resulted in faster alloy deposition and ~50% higher P content. The oxygen content in the coatings was sharply reduced by an order of magnitude, from ca. 20 at.% to 5 at.%.

Coatings with higher P contents (12 at.% and 16 at.%) were more noble than the low alloyed and the pure Fe sample and showed potentially better corrosion resistance. All layers with <16 at.% P were more or less inhomogeneous exhibiting pores that led to local element corrosion. Only the 16 at.% P sample exhibited a passive region. The measured composition after the corrosion test showed an almost doubled P content probably due to the formation of a phosphite or phosphate layer.

The increase of *H* hardness observed in the coatings from 6 at.% to 12 at.% P can be ascribed to grain boundary strengthening due to grain size reduction and to partial amorphization of the alloy as a result of P addition. A smoother friction force curve as a function of the scratch distance was observed for the Fe-16 at.% P coating. The abrupt drops observed in the friction force curves and in the friction coefficient curves can be ascribed to the inhomogeneous morphology.

**Author Contributions:** Conceptualization: N.K., W.H., and W.K.; Methodology: N.K. and W.K.; Validation: N.T., H.C., A.G., J.F., E.P., J.S., W.H., and W.K.; Formal Analysis: N.K. and W.K.; Investigation: N.K. and W.K.; Resources: W.H.; Writing—Original Draft Preparation: N.K. and W.K.; Writing—Review and Editing: N.K. and W.K.; Visualization: N.K. and W.K.; Supervision: W.K.; Project Administration: W.H.; Funding Acquisition: W.H.

**Funding:** The authors acknowledge funding by the HORIZON2020 SELECTA project (No. 642642), and partial financial support from the Generalitat de Catalunya (2017-SGR-292 project). J. Fornell and E. Pellicer are grateful to MINECO for the "Juan de la Cierva" (IJCI-2015-27030) and "Ramon y Cajal" (RYC-2012-10839) contracts, respectively.

**Acknowledgments:** Experimental assistance by M.B. Fernandez (IFW), M. Johne (IFW), V. Gman (Hirtenberger Engineered Surfaces GmbH), G. Quorri (Hirtenberger Engineered Surfaces GmbH), R. Mann (Hirtenberger Engineered Surfaces GmbH), and A. Nicolenco (Vilnius University) is gratefully acknowledged. Partial financial support from the MAT2017-86357-C3-1-R research project from the Spanish Ministerio de Economía y Competitividad (MINECO), cofinanced by the 'Fondo Europeo de Desarrollo Regional (FEDER), is also acknowledged.

**Conflicts of Interest:** The authors declare no conflict of interest. The funders had no role in the design of the study; in the collection, analyses, or interpretation of data; in the writing of the manuscript; or in the decision to publish the results.

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
