# Peer review of "Electrodeposition of Nanocrystalline Fe-P Coatings: Influence of Bath Temperature and Glycine Concentration on Structure, Mechanical and Corrosion Behavior"

_coatings, doi:10.3390/coatings9030189_

Reviewer 1 Report

In this work, different Fe-P coatings were electrodeposited under various temperature and glycine concentrations. The electrochemical and corrosion behaviour as well as mechanical properties of these coatings were investigated. The English writing of the manuscript (ms) is not enough good and it is not easy to understand.  The potentiodynamic section hasn’t been well presented and explained. However, from the technical point of view, there are some novelties in this work. The manuscript is not acceptable in its actual form and needs to major revision. 

Author Response

In this work, different Fe-P coatings were electrodeposited under various temperature and glycine concentrations. The electrochemical and corrosion behaviour as well as mechanical properties of these coatings were investigated. The English writing of the manuscript (ms) is not enough good and it is not easy to understand. The potentiodynamic section hasn’t been well presented and explained. However, from the technical point of view, there are some novelties in this work. The manuscript is not acceptable in its actual form and needs to major revision. Some of the encountered problems are mentioned as below:

General commentary: potentiodynamic section has been modified.

1.     Some of the important mistakes observed in the text are as followings:

-         The format of reference numbers is not uniform in all text. The authors used almost the ref. the number in form of exponent while in potentiodynamic section (page 3), the numbers have written directly in the text.
Commentary: the references have been amended.

-         Why H(gly) for glycerine? What does mean H in this term?
Commentary: some authors present glycine as H(gly). It shows that molecule of glycine could be easily deprotonated and increase the hydrogen evolution by the high concentration of the H+. The symbol for H(gly) was changed to Gly.

-         Line 132, page 3: “An increase of bath temperature a higher hydrogen evolution (first wave) and Fe-P deposition rate (Figure 1b)”. This sentence is not clear at all.
Commentary: text has been amended: “An increase of the bath temperature caused a higher hydrogen evolution (Figure 1b). 

-         Line 176, page 5: The authors have written: “The partial current densities for Fe decrease with temperature”. They should clarify with the increase or the decrease of temperature.
Commentary: Amended text: “The partial current densities for Fe decreased with the high temperature, whereas that of P and the hydrogen evolution increased. 

2.     In the potentioynamic section, line 133, the authors wrote: The polarization curves could be divided in two regions (Figure 1b). These regions should be described or showed clearly on the figures 1a and 1b.
Commentary: The Figures has been modified and text was added.

3.      A zoom around onset of cathodic peaks on cyclic voltammograms should be added to figure 2. The onset potentials are not clear in presented figure (Fig. 2).
Commentary: Figure2 has been split.

4.       Line 240, page 8: “Layers with a higher P content of 12 and 16 at% showed a more anodic Ucorr …” The coatings including higher P contents, show less negative potentials as compared to low P content coatings. Therefore, they are less anodic and more noble that the low P alloyed.
Commentary: More noble means more electropositive. Text has been amended according to common terms in electrochemical terminology.

5.      Line 246, page 8, the authors declared that: “Only the 16 at% P sample exhibited a passive region (ca. 200 mV …). According to the curves presented in Figure 7, all alloyed coatings (6, 12 and 16 at% show a passive region. How do you explain that?

Commentary: All curves reach under anodic conditions a wide plateau of the current density, but as this is relatively high, i.e. >1 mA/cm²  this must be rather interpreted as diffusion-limited dissolution current. On the other hand, the specific surface area of the electrodeposited layer may be much larger than the geometrical working electrode area (which is typically used for polished bulk material surfaces). This may explain the overall higher current density level (jcorr and janodic) than what would be expectable for bulk Fe in such a sulphuric acid solution. Altogether we would ascribe this anodic current density plateau as “pseudo-passive” region. This term is used for example for Mg corrosion and indicates that very low protective surface layers may be present which enable detectable material dissolution. While nearly all materials show after the jcorr minimum an immediate strong anodic current density increase to reach this “pseudo-passive” regime, only for  the Fe-15(16)P curve the anodic current density increases more gradually indicating a tendency for a beginning  “real passivity” before, at higher potentials, it also transfers into the other plateau regime.   

Text amended: ``All the coatings showed a passive region by the slow corrosion processes, but only the 16 at.% P sample exhibited a higher passive region in comparison to other coatings (ca. 200 mV anodic of Ucorr).``

6.      The discussions in the mechanical properties section (friction) were not supported by data presented in Fig. 9. In the line 284, page 10, the authors have written: “Above this value (normally 33 mN?), the Fe- 16 P sample yielded a lower slope and thus a lower friction coefficient (Figure 9a). Where the data of friction force vs. the normal force for these values have been presented?

Commentary: Figures and text have been amended.

Reviewer 2 Report

The article presented more indifferently. Sentences are stretched and long, sometimes tiresome to read — suggestion to the authors to construct more simple conceivable sentences and incorporate them into writing. Writing tone should be pertaining more active voice. Representation of graphs and tables are not up to the mark. Changes need to be incorporated. Also some serious issues regarding the reference presentation which I have mentioned in this review. Reference no. 11 says in publication. I am not sure whether it is accepted or not. If it is not accepted and under review, I think then authors should find some other reference.

Author Response

Review:

Summary: This article deals with the influence of bath temperature, glycine concentration on the structural, mechanical and corrosion behavior of nanocrystalline Fe-P coating on brass by electroplating. A detail electrochemical investigation carried out of the Fe-P alloy coupled with glycine bath at different temperatures together with corrosion and mechanical behavior. Low content of glycine to the electrolyte prompted a radical increment of the P content. Lower Fe-P deposition rate results in more inhomogenous porous coating compared to higher deposition rate. An increase in the P content from bringing about a decrease from nanocrystalline to “amorphous-like” surface. Coatings with higher P substance demonstrated a possibly better corrosion obstruction. Just the 16 at% P test displayed a passivated area and showed a nearly two times P content likely because of the development of a phosphate layer. Only the Fe-16 P test yielded a lower friction coefficient conceivably because of a phase changes.

Broad comments:

Strength: The nature and approach of the paper is unique and has a possible application in coatings.

Weakness: The article presented more indifferently. Sentences are stretched and long, sometimes tiresome to read — suggestion to the authors to construct more simple conceivable sentences and incorporate them into writing. Writing tone should be pertaining more active voice. Explicitly, one claim regarding coating thickness has not supported with shreds of evidence. Representation of graphs and tables are not up to the mark. Changes need to be incorporated. Also some serious issues regarding the reference presentation which I have mentioned in this review. Reference no. 11 says in publication. I am not sure whether it is accepted or not. If it is not accepted and under review I think then authors should find some other reference. Based on the significance of the work I would like to provide this article an opportunity to “reconsider after major revisions”. All comments under specific comments section are must address.

Specific Comments:

Line 22: “ a corrosion” to be only “ corrosion” and “was presented” passive voice change it.

Commentary: amended

Line 24 (25): “ was reduced” to be “ were reduced.”

Commentary: amended

Line 30: “ could be ascribed” to be “ could ascribe” ( passive voice to active voice).

Commentary: amended text: “The increase of hardness observed with increasing P content may be related to grain boundary strengthening.”

Line 31: “ were” to be “ was.”

Commentary: amended

Line 39: “ defects like grain boundaries, dislocations, etc.” to be changed to “ defects such as grain boundaries, dislocations.” (The use of etc. in formal writing is generally frowned upon. Consider rewriting the sentence.)

Commentary: amended text: “… owing to their single-phase nature related with the lack of structural defects like grain boundaries and dislocations.”

Line 40: Remove the “In particular,”. Start from “Fe-P...”

Commentary: amended

Line 43: “ e.g.” to be “,e.g.” (The abbreviation e.g. seems to be incorrectly punctuated. Consider changing the punctuation.)

Commentary: amended

Line 52: “ be well defined” (sounds like passive voice) changed to active voice.

Commentary: amended text: “Therefore, one has to define the experimental parameters and the electrolyte composition in order to obtain the targeted alloy composition and structure.”

Line 63: “was mostly pursued” passive voice change to active.

Commentary: text needs passive voice

Line 70: “were studied” again passive voice.

Commentary: text needs passive voice

Line 78: “were investigated” passive voice. Change it. “ In relation to” can be changed to “ concerning / about”.

Commentary: text needs passive voice

Line 80: “ were electroplated” again passive voice. Add “ an” before “aqueous”.

Commentary: text needs passive voice. Amended text: “Fe-P alloys were electroplated from an aqueous solution….”

Line 84: “with” should be “in”

Commentary: amended

Line 86: “was used” again passive voice.

Commentary: text needs passive voice

Line 88: “ are given” again passive voice.

Commentary: amended text: “All potential values are given vs. the SHE in the context of this paper.”

Line 91: “ was set” again passive voice.

Commentary: text needs passive voice

Line 92: Authors mention the thickness is 11 micron. How they measured it. It should be mentioned in the text. Did they use AFM? If show show enough evidences to bolter that thickness statement.

Commentary: amended text: “This thickness value and current efficiencies, h, were calculated based on weighing the total coating mass and an elemental weight analyses (EDX) of the electrodeposited alloys and Faraday’s law (30,31).”

Details of this EDX data based procedure are described in Ref. 31.

Line 94: “were morphologically and compositionally characterized” passive voice. “ with” to be changed to “by”.

Commentary: text needs passive voice. Text amended.

Line 97: “ was “ should be “is”.

Commentary: entire experimental text is in past time.

Line 99: “were also performed “ change to active voice.

Commentary: text needs passive voice.

Line 100: “were positioned” change to active voice.

Commentary: text needs passive voice.

Line 102: “ was contacted” again passive voice. “a copper” (The indefinite article, a, may be redundant when used with the uncountable noun tape in your sentence. Consider removing it.)

Commentary: text needs passive voice.

Line 103: should be “as a reference”.

Commentary: amended

Line 105: It appears that the noun OCP might combine better with an adjective other than respective. Consider rewriting this word pair or choosing a synonym for respective like an individual. Instead “ Each experiment was” consider using “Each experiment is”.

Commentary: amended

Line 106: Write three instead of 3.

Commentary: amended

Line 107: “were performed” again passive voice.

Commentary: text needs passive voice.

Line 108: replace the “ with a” to “with the”.

Commentary: general article needed in this case.

Line 111: “segment” should be “segments”. “ are reported” is a passive voice. Change it.

Commentary: text needs passive voice. Amended

Line 115: Write four instead of 4.

Commentary: amended

Line 116: “ was performed” to be changed to active voice.

Commentary: text needs passive voice.

Line 120: “were considered” to be changed to active voice.

Commentary: text needs passive voice.

Line 121: “in a similar manner” may be considered wordy. Consider changing the wording. Replace with Similarly/ correspondingly.

Commentary: amended.

Line 125: “were measured” again passive voice. Change it.

Commentary: text needs passive voice.

Line 126: Change “gly”to “Gly”. Change it for all.

Commentary: amended.

Line  127: “ represent” to be “represents”.

Commentary: singular.

Line  128: “ is shifted” is again passive.

Commentary: amended.

Line 129: “This” It may be unclear who or what This refers to. Consider rewriting the sentence to remove the unclear reference. Also “due to the fact that” is maybe wordy. Consider changing the wording by “since”.

Commentary: Amended text: “This deposition rate decrease is due to the fact that the addition of Gly increases the concentration of adsorbed H+….”

Again in “ addition” appears that an article is missing before the word addition. Consider adding the article. Like “the addition”.

Commentary: amended.

Line 131: Instead of “effective” use “active”. “of” should be “in.”

Commentary: amended.

Line 132: “ hydrogen evolution (first wave) and Fe-P deposition rate (Figure 1b).” This appears to be an incomplete sentence. Consider rewriting the sentence or connecting the fragment with another sentence.

Commentary: Amended text: “An increase of the bath temperature caused a higher hydrogen evolution (Figure 1b).”

Line 133: “ be divided” again passive voice. Change it. Consider “in” to “ into”.

Commentary: Amended text: “The polarization curves were divided into two regions (by a dashed line).”

Line 135: “ coexist” change to be “ coexists”.

Commentary: plural.

Line 139: Figure 1 a and b both are misaligned. Also, can authors increase the upper limit of X-axis more than zero so that the proton reduction starting point is visible? It was more than zero, don’t understand this comment.

Commentary: figures amended.

As mentioned in the text according to figure 1a it is at -0.5V. But in the figure, it is showing very near about -0.6V. it is near 0.5 Can authors clarify why it is -0.5V? Don’t understand this comment, but probably it has to be described more in detail!, Can authors point out the hydrogen evolution peaks in figure 2b for those respective three graphs? Please mention the direction of the curve.

Commentary: figures amended.

Amended text:

“The polarization curves were divided into two regions (by a dashed line). The reaction into the first region is mainly related to hydrogen evolution. In the second region, hydrogen evolution and Fe-P deposition coexisted. Figure 1a shows the influence of the glycine, Gly, content in the electrolyte at 20 °C. The current density waves starting at ca. -0.5 V represent the proton reduction without Fe-P electrodeposition. The study of the Fe-P deposition potential with varying Gly concentration were performed by cyclic voltammetry (Figure 2). The potential transients were starting from the individual OCP to the negative direction with varied reversal potentials. The first appearance of the anodic striping peak correlates to the Fe-P deposition potential that is caused by second current density wave. Cyclic voltammetry shows that the onset of the cathodic Fe-P deposition shifts to more negative potentials with increasing Gly concentration. Table 1 shows an average potential of the second current density wave. The addition of Gly decreased the deposition rate caused by the competing hydrogen evolution. Therefore, the high concentration of adsorbed hydrogen species (H+, compare (33)) decreased the active surface area for metal deposition.”

Line 143: Figure 2 can they split these three graphs in a, b & c. Because it is not possible to recognize each graph, only the reverse direction given in the arrow. No forward direction arrow is there in solid and dashed curves. It is not clear what both curves mean? Clarify it on the figure caption.

Commentary: figure 2 has been split

The green curve has more crossovers than others. Is it due to more glycine deposition on the surface or adsorbed hydrogen? They should explain this in the text. If I compare the green curve (for 0.64M Gly) with Figure 1a the oxygen reduction peak is not recognized. Though it is understandable in acidic medium, can they clarify why such differences is assuming that both are same samples?

Commentary: The addition of Gly decreased the deposition rate caused by the competitive increasing of the hydrogen evolution (not oxygen reduction). Therefore, the high concentration of adsorbed H+, decreases the active surface area for metal deposition. It should be be taken into account that the current density scale of Fig. 2c contrasts to that of Fig. 1a and 1b, when comparing the cathodic current waves.

Line 144: Possibly miswritten “ in dependence”. I guess it should be “concerning/ to.”

Commentary. Amended.

Line 146: “ are indicated” again passive voice.

Commentary: amended

Line 147 to 149: The potential mentioned in figure 2 and table 1 are not the same for 0M Gly, 0.11M Gly. Did author mention only average values of potentials corresponding to currents in table 1? Then they need to provide the standard deviation too with those potential values.

Commentary: the average potential of the second wave was chosen (see modified Figure 2). Text was added: ”Table 1 shows an average potential of the second current density wave”.

Line 150: The “dependency of “ should be “dependency on.”

Commentary: amended

Line 152: It appears that Obviously may be unnecessary in this sentence. Consider removing it.

Commentary: amended

Line 153: “Further concentration increase led to a slight P increase.” Can they explain it why?

Commentary: amended text

“Further concentration increase led to a slight P increase, due to its complexation with iron. The formed complex lead to a deposition rate decrease of Fe resulting in a higher codeposition rate of P.”

Line 155: Instead of “strongly” use sharply. It seems that the whole sentence would turn in a passive voice. Try to rewrite in an active voice.

Commentray. amended

Line 158: Should be “ in the air.” Instead of “ following” using “after electrodeposition” would be more lucid to understand.

Commentray. amended

Line 160: Figure 3. Why authors have not shown any P or O content for 0.32M and 0.43M of Gly, but they have mentioned that on table 1 for onset potentials. Why it is so, can they represent those graphs in more color format? Also, use “The” before dependency.

Commentray. amended

Line 163: “ be evaluated” again passive voice. Change it.

Commentary: current density values was evaluated. Text needs therefore passive voice.

Line 164: First, according to reference 33 what authors have mentioned it only have FTIR and electrochemistry data. They never explicitly mentioned this formula in the paper.

Commentary: Reference 33 was not presented in the text. The reference is [A. Nicolenco, N. Tsyntsaru, and H. Cesiulis, “Fe (III)-Based Ammonia-Free Bath for Electrodeposition of Fe-W Alloys,” J. Electrochem. Soc., vol. 164, no. 9, pp. D590–D596, 2017]. The authors performed the same calculation. They were performed with the formula shown in the partial current section.

Mention the name of the formula how EDX data can determine weight in (g/cm2). EDX determines the atomic composition of the specimen.

Commentary: the weight percentage of the elements could thus be determined.

It does not give chemical information (e.g., oxidation state, chemical bonds). For quantitative analysis, EDX is not suitable for light elements (like O). It can detect the presence of oxygen, but the quantification is tricky.

Commentary: the electron valencies were assumed for Fe are 2 and for P is 1.

Can they explain it how they estimated the mass from EDX? It can still predict the concentration in g/mol with K ratios if the standard concentration known with ZAF factors. Please explain it with a proper formula.

Commentary: the formula has been added.

Line 170: It is unclear about what potentials they calculated the current densities of required for Fe, P. Mention it in the table2 caption it is calculated regarding table 1 potentials (I guess). Also please identify Hydrogen evolution peaks in figure 2.

Commentary: The presented data in Table 2 were calculated according to the Equations (1) and (2).

The calculations were performed according to a known current density (and time) which was applied for electrodeposition for a layer thickness of 10µm.

Line 172: replace “main” with a primary.

Commentary: amended

Line 173: remove the comma after evolution.

Commentary: amended

Line 174: “ were used” again passive voice.

Commentary: text needs passive voice.

Line 175: “ were compared.” again passive voice.

Commentary: text needs passive voice.

Line 180: Put a comma after also. Also, “was already indicated” in passive voice change it.

Commentary: text needs passive voice.

Line 183: “ The technical praxis of electroplating is dominated by galvanostatic processes” can be changed to “ Galvanostatic processes dominate the technical praxis of electroplating.”

Commentary: amended

Line 183 to 187: This is a complex conjugated sentence. Can authors break it in multiple sentences with active voice. “mention what is η.

Commentary: text amended: “Therefore, the results of the respective current efficiency,h, of the Fe-P alloy deposition together with the P layer content (Figure 4a) and the deposition rate (Figure 4b) are presented versus the applied current density at 60°.”

Line 190: It appears that starts out creates a tautology. Consider removing it.

Commentary: amended

Line 197: Use The dependency.

Commentary: amended

Line 206: “ are controlled” is in a passive voice. Change it.

Commentary: amended text: “ The competition between hydrogen bubble evolution and the layer growth control the morphology.”

Line 210: “is reduced” is in a passive voice.

Commentary: text needs passive voice

Line 211: It may be unclear who or what This refers to. Consider rewriting the sentence to remove the unclear reference.

Commentary: amended

Line 212: pace “in” before diameter.

Commentary: amended

Line 214: replace “ and” by “ also,”

Commentary: text needs “and”

Line 228: “was performed” is in a passive voice.

Commentary: text needs passive voice.

Line 234: Figure 7. shows cyclic polarization curves. The curves appear not fully shown in figure 7. Authors need to show the full polarization curves showing forward and reverse direction. Prefer to split it into two pics one having full polarization curves and other one showing the Tafel slope region.

Commentary: Curves are now shown in full potential range. Figures show a potentiodynamic polarization curves, not cyclic polarization curves. How the measurements were performed is described in the experimental part.

Line 238: Can they incorporate corrosion resistance in table 3.??

Commentary: amended

Line 241: “ more noble” can be nobler. In what context it is more noble???However, can they stay more quantitative and specific in this context rather than using the nobler term?

Commentary: “More noble” means practically more “electropositive” in respect to the observed corrosion potential (a common terminology in electrochemical corrosion research). In the text “More noble” is now replaced by “more electropositive”.

Line 244: It appears that Actually, may be unnecessary in this sentence. Consider removing it. Also, “ were almost completely dissolved” is in a passive voice.

Commentary: text amended. Text needs passive voice.

Line 247: “was reduced” is in a passive voice.

Commentary: text needs passive voice

Line 253: “ Actually, elevated” can be replaced by “ High.”

Commentary: amended

Line 254: Put a comma after “Also,”.

Commentary: amended

Line 268: “larger” can be replaced by “substantial.”

Commentary: amended

Line 270: It should be “e.g.,”

Commentary: amended

Line 271: “ are” should be “is”.

Commentary: Has to be plural.

Line 273: It should be “i.e.,”

Commentary: amended

Line 280: “was studied” is a passive voice.

Commentary: text needs a passive voice

Line 283: “ is” should be “ are”.

Commentary: text needs a passive voice

Line 287: “was observed” again a passive voice.

Commentary: text needs a passive voice

Line 290: Replace the “ a corrosion” by “corrosion”. Also, “was presented” is in the passive voice.

Commentary: amended. Text needs passive voice

Line 292: Omit empty phrases like “ Obiviously.”

Commentary: amended

Line 293-294: “The concentration increases of glycine decreased the partial currents of Fe and P and increased that of the proton discharge which supported the co-deposition of P.” Can they rewrite this sentence by splitting it into simple sentences?

Commentary: text amended:”The increase of the Gly concentration decreased the partial currents of Fe and P but increased the hydrogen evolution which supported the co-deposition of P”

Line 295: “The observed layer morphologies were controlled by the competition between hydrogen bubble” can be rewritten to make it more understandable.

Commentary: text amended: “The competition between hydrogen bubble evolution and the layer growth control the morphology.”

Line 297: “were observed” again in a passive voice.

Commentary: amended

Line 301: Change “ strongly” to sharply.

Commentary: amended

Line 303: Change the “more noble” term.

Commentary: more noble is correct

Line 305 to 307: “Only the 16 at% P sample exhibited a passive region and showed an almost doubled P content probably due to the formation of a phosphite or phosphate layer.” Can authors split this sentence and rewrite it more lucid.

Commentary: text amended: “All the coatings showed a passive region by the slow corrosion processes, but only the 16 at.% P samples exhibited a higher passive region in comparison to other coatings (ca. 200 mV anodic of Ucorr).”

Reference:

All the references were corrected according to journal style.

Line 358: Reference 13 the article citation should be: “Cserei, A., Kuzmann, E., Pöppl, L. and Vértes, A., 1994. Study of the crystallization kinetics in amorphous Fe83P17 alloy. Journal of Radioanalytical and nuclear chemistry, 187(1), pp.33-45.”

Commentary: amended

line 374: Reference 20 author names are not in the right manner. It should be “Cuculić, V., Pižeta, I. and Branica, M., 2005. Voltammetric determination of stability constants of iron (III)–glycine complexes in water solution. Journal of electroanalytical chemistry, 583(1), pp.140-147.”

Commentary: amended

Round  2

Reviewer 1 Report

The manuscript is acceptable in the present form.

Author Response

Authors thank the reviewer for his assistance!

Reviewer 2 Report

I am happy with the explanations. Please do incorporate the edits I provided in my review. It looks good.

Author Response

The authors thank the reviewer for his valuable suggestion, which have been considered in the amended text.
